# Effect of Egg White Protein and Soy Protein Isolate Addition on Nutritional Properties and In-Vitro Digestibility of Gluten-Free Pasta Based on Banana Flour

**DOI:** 10.3390/foods9050589

**Published:** 2020-05-06

**Authors:** Adetiya Rachman, Margaret A. Brennan, James Morton, Charles S. Brennan

**Affiliations:** 1Department of Wine, Food and Molecular Biosciences, Lincoln University, Christchurch 7647, New Zealand; Ade.Rachman@lincolnuni.ac.nz (A.R.); margaret.brennan@lincoln.ac.nz (M.A.B.); james.morton@lincoln.ac.nz (J.M.); 2Riddet Research Institute, Palmerston North 4442, New Zealand; 3Indonesia Institute for Agricultural Research and Development, Jakarta 12540, Indonesia

**Keywords:** banana pasta, soy protein isolate, egg white protein, digestibility, amino acid

## Abstract

The effects of egg white protein and soy protein isolate addition on the nutritional and digestibility of gluten-free pasta based on banana flour were studied. The level of protein additions (soy protein or egg white protein) were 0, 5, 10 and 15% of banana flour (*w*/*w*). Pasta made from 100% durum wheat semolina was used as a control. Soy protein isolate inclusion into banana pasta increased total phenolic content (TPC) and antioxidant capacities, while egg white protein decreased the TPC and antioxidant capacities with the increasing level of addition. Starch digestibility was affected by the type of protein addition. Egg white protein lowered starch digestibility compared to soy protein isolate. Protein inclusion in banana pasta also altered protein digestibility, amino acid profiles and protein digestibility-corrected amino acid score (PDCAAS). Soy protein isolate increased protein digestibility of gluten-free pasta compared to egg white protein. Protein enrichment gave better amino acid profiles of banana pasta compared to semolina pasta with egg white protein and performed a better PDCAAS compared to soy protein isolate. These results showed that soy protein isolate and egg white protein addition enhanced nutritional qualities and digestibility properties of gluten-free banana pasta.

## 1. Introduction

The effort to provide suitable pasta for celiac and gluten intolerance-related patients has led to the production of gluten-free (GF) pasta. There are many varieties of GF pasta made from rice and other GF flours available, but these products often have poor cooking quality and technological difficulties compared to conventional wheat pasta as well as having an inferior nutritional quality especially with regard to minerals and bioactive compounds [1,2]. Commercial GF products also often perceived as unattractive from a consumer perspective as unhealthy products and with lots of artificial ingredients [3].

One GF flour that has great potential to be developed as a GF pasta ingredient because of its nutritional properties is banana flour. Banana flour is derived from unripe bananas which have a high phenolic content and high antioxidant capacity [4]. Banana flour is mainly composed of starch, with a relatively low protein content, but the high level of bioactive ingredient in banana makes it useful as a functional foods ingredient. The bioactive compounds in bananas remain high and active after processing into banana flour which makes it a good nutritional source for the food industry [5].

Banana flour has been applied in either wheat-based or GF pasta manufacture as an alternative base material, improving the pasta and making it comparable with wheat only counterparts [6,7,8,9,10,11]. It also has been reported in previous work that banana flour has great potential to be developed as gluten free pasta due to its physicochemical and digestibility properties [12].

Some of the research into banana pasta stated the nutritional values of pasta products, but none investigated how the ingredients affected the nutritional and digestibility properties of the final product. Sarawong at al. [6] reported a high content of resistant starch in banana pasta but did not observe any phenolic content or antioxidant activities. Ovando-Martinez et al. [13] reported phenolic content, antioxidant activities and starch digestibility in pasta made from up to 45% banana flour, but still used semolina as a base ingredient. There is limited research discussing the nutritional aspects of gluten-free pasta based on banana flour.

Most commercial GF pasta use hydrocolloids and emulsifier to improve cooking quality which led to the consumer association with artificial flavour [14,15]. Another functional additive that has been proven to improve the quality of GF product is protein source, such as egg white protein and soy protein isolate [15]. Protein enrichment with egg protein has successfully improved the cooking quality and textural properties of banana pasta [6,9]. It has also been reported that soy protein isolate addition enhanced gluten-free rice spaghetti quality [16]. Previous research has been carried out to incorporate soy protein isolate and egg white protein into banana flour and has improved the cooking quality and textural properties of banana pasta [17]. The aim of this study was to investigate the effects of the protein type (egg white protein and soy protein) and the level of protein addition on nutritional qualities (total phenolic content, antioxidant capacities and amino acid profiles) and digestibility properties (starch digestibility and protein digestibility) of gluten-free pasta based on banana flour.

## 2. Materials and Methods

### 2.1. Materials

Banana flour was obtained from Food Compass Limited (Auckland, New Zealand). Soy protein isolate (with a protein content of 91%) was obtained from Bulk Powder Limited (Braeside, Australia). Egg white protein powder (80% protein content) was bought from Nothing Naughty Limited (Tauranga, New Zealand). Durum semolina (from Sun Valley, Auckland, New Zealand) was used as a control.

### 2.2. Pasta Preparation

Banana flour was mixed with soy protein isolate or egg white protein powder in different levels: 0%, 5%, 10% and 15% addition of flour (*w*/*w*). Semolina was also used to make control pasta. A pasta-making machine fitted with a spaghetti die (2.25 mm diameter of die hole with 20 holes, model: MPF15N235M, manufactured by Fimar, Villa Verucchio, Ravenna, Italy) was used to produce the pasta.

All dry components (300 g) were mixed in the pasta maker for 4 min. The pH of the mixture was kept at 7 for comparison purposes with the control pasta. Boiling water was added at 70% (*w*/*v*) and mixed for 20 min based on previous research [12]. Semolina pasta as a control was prepared by adding 30% water at 41 °C [18]. Pasta dough was then extruded through the spaghetti die 2.25 mm diameter. Pasta was cooked in boiling water (100 °C) based on its optimum cooking time prior to analysis [17].

### 2.3. Extraction of Sample for Total Phenolic and Antioxidant Capacity

Samples for analysis of total phenols and antioxidant capacity were obtained as previously described by Hossain et al. [19] with slight modification. Cooked pasta was placed into 60 °C oven air overnight and grinded to make a pasta powder. Cooked pasta powder (2 g) was mixed with 20 mL of 70% methanol, kept in the multi-stirrer overnight at room temperature (approximately 25 °C) and then centrifuged at 3000 g (2500 rpm) for 10 min. The supernatant was collected into 15 mL plastic tubes and kept at −20 °C until analysis.

### 2.4. Total Phenolic Content

The total phenolic content (TPC) of the sample extracts was determined spectrophotometrically using Folin-Ciocalteau’s (F-C) reagent according to the method described by Lim and Murtijaya [20] and Khanizadeh et al. [21] with slight modifications. The sample (500 µL) was added to the test tubes followed by 2.5 mL of 0.2 mol/L Folin-Ciocalteau reagent and 2.0 mL of sodium carbonate (7.5 g/100 mL). The contents of the tubes were mixed thoroughly and stored in the dark for 2 h before the absorbance was measured at 760 nm using VWR V-1200 Spectrophotometer (VWR International Co., Pennsylvania, USA). Gallic acid (0–200 µg) in methanol was prepared for standard. TPC was expressed as mg gallic acid equivalents (GAE) per 100 g of fresh material.

### 2.5. Ferric Reducing/Antioxidant Power (FRAP)

FRAP was assessed according to Khanizadeh et al. [21] with slight modifications. A fresh working solution of FRAP reagent was prepared each time by mixing acetate buffer (300 μM, pH 3.6), a solution of 10 mM TPTZ in 40 mM HCL, and 20 mM FeCl_3_·7H_2_O at 10:1:1 (*v*/*v*/*v*). A standard of iron (II) sulphate (FeSO_4_·7H_2_O) (0–200 μmol) or sample extract (250 μL) was added to 2.5 mL of the FRAP reagent and the absorbance at 593 nm was recorded immediately after the addition of the sample and again after 2 h incubation at 37 °C [22]. The results were expressed as μmol Fe^2+^/g sample.

### 2.6. ABTS (2,2′-azino-bis(3-ethylbenzothiazoline-6-sulfonic acid)) Radical Scavenging Capacity

The ABTS radical scavenging assay was based on the method of Cai et al. [23]. ABTS^•+^ was prepared by reacting colourless ABTS stock solution (7 mM in water) with 2.45 nM potassium persulfate and allowing the reaction to stand for 16 hours in the dark at room temperature. On the day of analysis, the ABTS^•+^ solution was diluted with PBS (pH 7.4) to an absorbance of 0.70 (±0.02) at 734 nm and 3 mL transferred to a cuvette. After the addition of 300 μL Trolox (0–200 μmol) or sample extract, the mixture was well mixed, allowed to stand for 6 min and the absorbance was read at 734 nm. The results were expressed as Trolox equivalents (TE).

### 2.7. Starch Digestibility

Cooked pasta (5 g) were cut with a knife to obtain a 2–5 mm size. The reducing sugars released over 120 min were evaluated for each pasta type as described by Foschia et al. [18].

### 2.8. Protein Content for In-Vitro Protein Digestibility Determination

Protein content was determined by using element analyser model Vario MAX CN to determine percentage of total nitrogen. The value of total nitrogen content was multiplied by 6.25 to get crude protein content value.

### 2.9. In-Vitro Protein Digestibility

In-vitro protein digestibility of cooked pasta was performed using a multi-enzyme technique following the method used by Desai et al. [24]. Protein was suspended in distilled water (6.25 mg of protein/mL) and adjusted to pH 8 using 0.1 N HCL and/or 0.1 N NaOH, and then placed on magnetic heating stirring block at 37 °C. A multi-enzyme solution (1.6 mg/mL Trypsin, 3.1 mg/mL chymotrypsin and 1.3 mg/mL protease) was kept in an ice bath. Five mL of this was added to the protein suspension at 37 °C. Decreases in pH were measured every minute for a period of 10 min using a digital pH meter (S20 Seven Easy™, Mettler Toledo, USA). The percentage protein digestibility (Y) was calculated by using the formula mentioned in the previous paper [24]:Y = 210.46 − 18.10 X(1)
where X represents the change in pH after 10 min.

### 2.10. Amino Acid Profiles, Amino Acid Score (AAS) and Protein Digestibility-Corrected Amino Acid Score (PDCAAS)

Amino acid profiles were determined using Agilent 1100 series (Agilent Technologies, Walbronn, Germany) high-performance liquid chromatography following the methodology used by Heems et al. [25]. The sample was hydrolyzed with 6 N hydrochloric acid in an oven at 110 °C for 20 h before being used. The HPLC column was 150 × 4.6 mm, C18, 3uACE-111-1546, (Winlab, Glasgow, Scotland) and used for the separation of amino acid. The mode of operation was flow rate (0.7 mL/min) maintained at 40 °C. Primary amino acids were detected using O-phethaldialdehyde (OPA) as a fluorescence derivative reagent, while secondary amino acids used 9-fluorenylmethyl chloroformate (FMOC). The fluorescence detector used an excitation of 335 nm and emission of 440 nm for primary amino acids. The detector was switched at 22 min to excitation 260 nm, emission 315 nm to detect secondary amino acids such as proline. The amino acid results are expressed in mg of amino acids per g protein. Amino acid scores were calculated by dividing the amino acid content of the sample (mg/g protein) by the suggested reference pattern of amino acid requirements (mg/g protein) for pre-school children (1–2 years old) for nine essential amino acids plus tyrosine and cysteine as follows: histidine—18, isoleucine—31, leucine—63, lysine—52, methionine + cysteine—25, phenylalanine + tyrosine—46, threonine—27, and valine—41 [26]. The protein digestibility corrected amino acid score (PDCAAS) was calculated by multiplying the corresponding protein digestibility percentage by the lowest AAS value. PDCAAS values exceed 1.00 were truncated to 1 [27].

### 2.11. Statistical Analysis

All experiments were conducted in triplicate. The analysis of variance (ANOVA) was conducted in Minitab statistical software (Minitab 18, Minitab Pty Ltd., Sydney, Australia). Statistical differences in all nutritional characteristics were determined by one-way ANOVA for all treatments and general linear model analysis of variance (ANOVA) with 2 factors (type of protein and level of protein) with semolina pasta excluded. Tukey’s comparison test (*p* < 0.05) was performed to determine significant difference between treatments.

## 3. Results and Discussion

### 3.1. Total Phenolic Content (TPC)

Total phenolic content (TPC) of raw materials and pasta samples are shown in Table 1. Banana flour has higher TPC compared to semolina flour, while soy protein isolate contains more TPC compared to egg white protein. These initial statuses led to higher TPC in banana pasta compared to semolina pasta, while addition of soy protein isolate gave a higher TPC compared to egg white protein enrichment. Choo and Aziz [8] made noodles incorporating 30% banana flour into wheat-based noodles and found significant increase in value of TPC. Noodles with 30% banana flour had more than triple TPC content compared to 100% wheat noodle as a control (28.6 and 90.4 mg GAE/100 g respectively).

High TPC in soy protein isolate also contributed to enhancing TPC of banana pasta. The TPC of banana pasta increased by increasing fortification level of soy protein isolate. Soy protein isolate alongside with other soy products has been reported to contain high total phenolic content ranging from 104.3 to 243.1 mg GAE/100 g [28]. Different effects were found with egg white protein inclusion; the level of egg white protein did not give significantly different values of TPC. There is limited research which discuss the effect of protein addition on total phenolic content in either wheat-based pasta or gluten-free pasta.

### 3.2. Antioxidant Capacities

Antioxidant activities including ferric reducing/antioxidant power (FRAP) and ABTS radical scavenging capacity of raw materials and pasta sample can be seen in Table 1. Similar to the TPC content, banana flour has higher FRAP and ABTS values compared to semolina pasta which then provided higher antioxidant capacities in banana pasta compared to semolina pasta. The higher ABTS values in banana pasta compared to semolina pasta meet agreement with Ovando-Martinez et al. [13] who carried out a study on undigestible carbohydrate of semolina pasta supplemented with banana flour. They found that the ABTS value of semolina pasta increased with increasing level of banana flour replacement. Semolina pasta had an ABTS value of 0.55 µmol Trolox equivalent/g and increased up to 0.89 µmol Trolox equivalent/g with 45% banana flour inclusion.

Soy protein isolate addition provided higher FRAP and ABTS values compared to egg white protein. A higher soy protein level gave higher antioxidant capacities especially for ABTS values. On the contrary, higher egg white protein incorporation in banana pasta reduced antioxidant activities. This can be explained by lower FRAP and ABTS values in egg white protein, while soy protein isolate has high value similar to banana flour. Soy protein utilization in many food formulations has been established because of its functional and nutritional properties [29]. Soy protein and egg white protein have also been utilised in GF products [30,31], but none of the previous studies examined the effect of the protein inclusion on nutritional properties of the GF products.

### 3.3. Starch Digestibility

The effect of protein incorporation on starch digestibility can be seen in Figure 1. Egg white protein fortification gave a lower value of area under curve (iAUC) compared to banana pasta with soy protein isolate addition. Banana pasta with egg white inclusion also has a lower iAUC value compared to unfortified banana pasta; however, this is not significantly different.

Some research projects have reported that protein inclusion in pasta based on semolina flour arrived at lower in-vitro starch digestibility properties compared to semolina pasta [24,32,33]. Numerous studies have been conducted to illustrate how starch digestibility is decreased by incorporating more fibre into wheat-based pasta [8,34,35,36]. Unfortunately, there is no published research that addresses the effects of protein inclusion on digestibility properties of gluten-free pasta. Hager et al. [37] made a comparison of egg pasta made from oat, teff and wheat flours and found that starch digestibility varies according to the pasta composition. Oat based-pasta with a lower egg white powder proportion (9.7%) was reported have a lower glucose released compared to teff based-pasta with higher egg white powder (11.0%). It has also been noted that the protein content of teff flour was higher compared to oat flour (12.8% and 6.9%, respectively), while both flour have similar fibre content (4.5% and 4.1%, respectively). This result explains how a higher protein content in the food matrix may lead to different effect in starch digestibility properties.

The amount of protein is not the only aspect to consider, but also the quality of protein itself [37]. Oat protein has a higher lysine content that makes it superior compared to other cereals [38], and the differences in protein composition may be why the addition of different types of protein in the same amount may give different effects on starch digestibility. Egg white protein seems to have superior quality compared to soy protein isolate in creating a protein-starch network in banana pasta which alters the starch digestion during the enzymatic reaction. A better quality of protein in the egg white builds a stronger banana pasta structure which can also be illustrated by superior cooking quality and texture properties of enriched banana pasta compared to soy protein isolate inclusion in previous work (Rachman et al., 2019). Shen et al. [39] studied amino acid enrichment in bread and found the addition of lysine, alanine or glycine improved the textural quality of white bread, implying that protein quality may have an effect on dough structures in food matrices.

### 3.4. Protein Digestibility

Protein digestion was observed by pH change over time during multi-enzymes (trypsin, chymotrypsin and protease) incubation of pasta samples which can be seen in Figure 2. Protein is digested into amino acids and peptides resulting in pH drop. Multi-enzymes break the protein solution into carboxyl (-COO^-^) and amino (-NH_3_^+^) groups. Free amino groups deionize in water and protons (H^+^) are liberated at neutral pH (8.0). The free H^+^ released into the solution decreased in pH value. The phenolic present in pasta could react with amino acid creating protein cross-link that inhibits further enzymatic protein degradation [24].

Protein content, protein digestibility and protein availability of pasta samples are shown in Table 2. Banana pasta without protein enrichment has the lowest protein digestibility due to its lack of protein content. Semolina pasta had the highest protein digestibility among all samples because of its high protein content. The effect of protein addition to the banana pasta increased protein digestibility with increasing fortification. Soy protein isolate addition gave higher protein digestibility compared to egg white protein. This result agrees with previous research which illustrated that protein digestibility of some commercial protein supplements including soy protein isolate and egg white protein powder varied greatly. For instance, it has been found that soy protein isolate had higher protein digestibility (89.2%) compared to egg white protein powder (81.5%) [36]. Interestingly, enriched banana pasta with similar or higher protein content (11.39–13.89%) compared to semolina pasta (12.26%) had a lower protein digestibility. Simonato et al. [37] observed that protein digestibility of pasta made from three types of wheat flour was not affected by protein content. The lower protein digestibility might be caused by high antinutrient content in banana pasta especially the phenolic content and soluble fibre, both of which can inhibit protein digestion during enzymatic reactions. Anti-nutrient activities against protein digestibility were also suggested by other research into protein digestibility in pasta [20,29,38].

### 3.5. Amino Acid Profiles and Protein Digestibility Corrected Amino Acid Scores (PDCAAS)

Amino acid profiles (mg per g protein) of the pasta samples can be seen in Table 3. The level of protein addition and the type of protein altered amino acid composition in banana pasta. Banana pasta had poor essential amino acid content, e.g., methionine, cysteine, and isoleucine, compared to semolina pasta. The addition of egg white protein or soy protein isolate enhanced the amino acid composition to become better balanced for daily requirements than semolina pasta. Egg white protein gave a better ratio of essential amino acid to total amino acid than soy protein isolate inclusion. Table 3 shows the comparison of amino acid profiles in the pasta samples with the recommendation of amino acid required for daily adult intake by WHO, FAO and UNU [26].

Amino acid scores and protein digestibility-corrected amino acid scores can be seen in Table 4. Semolina pasta had several limiting EAA (Essential Amino Acid) compared to WHO reference, with valine as the most limiting EAA. Banana pasta had most limiting EAA and showed lack of some EAA content: valine, isoleucine and cysteine + methionine, corresponding with WHO recommendation. Egg white protein inclusion in banana pasta gave a better AAS value compared with soy protein isolate and, with a level of minimum 10% egg white protein fortification, fulfilled AA recommendation for daily adult intake showed by no limiting AAS (all values > 1).

Protein digestibility-corrected amino acid score of cooked pasta samples ranged between 0.07 to 1.00 (Table 4). Banana pasta sample had the lowest value because of its limiting AAS and low in-vitro protein digestibility. Interestingly, all enriched banana pasta had higher PDCAAS value compared to semolina pasta while this pasta control had highest in-vitro protein digestibility among all the pasta samples. This result showed that enriched banana pasta performed as a better protein quality source compared to semolina pasta. Egg white protein also showed as a better protein quality compared to soy protein isolate, forming a higher PDCAAS values and, at level 15% egg white enrichment, exhibited the maximum value of PDCAAS. Egg white protein has been used as a protein quality standard and its PDCAAS value has similar optimal value as milk protein [40,41].

## 4. Conclusions

The addition of egg white protein and soy protein isolate into banana pasta altered the total phenolic content and antioxidant capacities compared to semolina pasta and pure banana pasta. Soy protein isolate increased TPC, ferric reducing/antioxidant power and ABTS radical scavenging capacity of banana pasta by the increasing level of fortification. On the contrary, egg white protein lowered phenolic content and antioxidant capacity values with increasing egg white protein addition.

Glycaemic properties of banana pasta were also affected by the type of protein addition. Egg white protein inhibited reducing sugars released in fortified banana pasta, while soy protein isolate did not have significant impact on starch digestion properties. Protein enrichment in banana pasta also altered protein digestibility. Protein digestibility increased along with the increased protein inclusion in banana pasta. Amino acid profiles of enriched banana pasta also enhanced and met daily requirement for adult human with optimum protein digestibility-corrected amino acid score. These results show that soy protein isolate and egg white protein addition into gluten-free pasta formulation based on banana flour improved the nutritional and digestibility properties of banana pasta better than those of semolina pasta.

## Figures and Tables

**Figure 1 foods-09-00589-f001:**
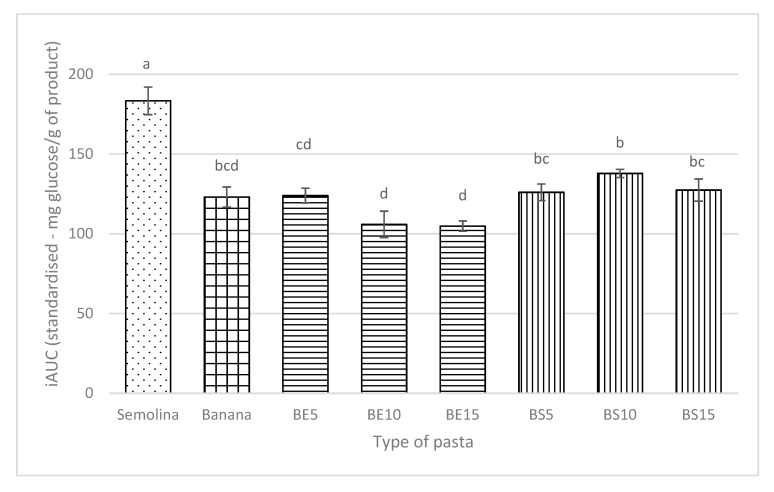
Area under curve (iAUC) values of banana pasta and banana pasta with protein addition (BE5, BE10, BE15 = Banana pasta with 5, 10, 15 g egg white protein/100 g banana flour, BS5, BS10, BS15 = Banana pasta with 15 g soy protein/100 g banana flour) compared to semolina pasta as a control. Different letters above the bars indicate values which are significantly different from each other (*p* > 0.05), according to Tukey’s test.

**Figure 2 foods-09-00589-f002:**
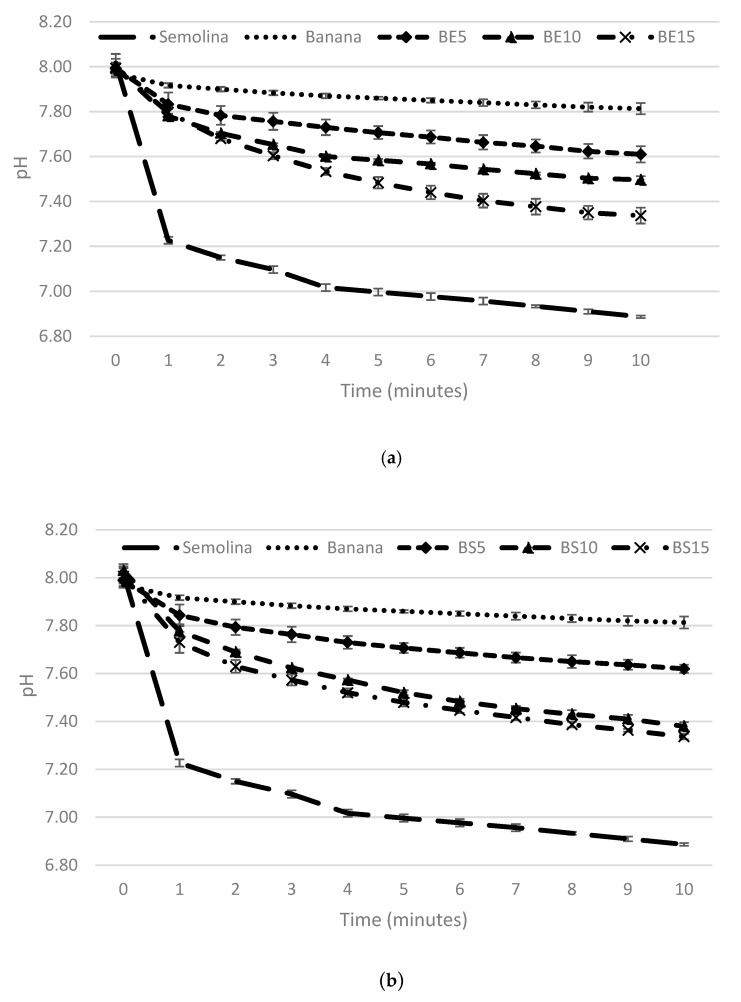
The pH change over time during multi-enzymes (trypsin, chymotrypsin and protease) incubation of semolina, banana pasta and banana pasta with protein addition: (**a**) BE5, BE10, BE15 = Banana pasta with 5, 10, 15 g egg white protein/100 g banana flour; (**b**) BS5, BS10, BS15 = Banana pasta with 15 g soy protein/100 g banana flour.

**Table 1 foods-09-00589-t001:** Total phenolic content and antioxidant capacities.

Formulation	TPC (mg GAE/100 g)	FRAP (mmol/100 g)	ABTS (mmol/100 g)
*Raw materials*			
Semolina flour	73.80 ± 0.78	0.15 ± 0.02	0.67 ± 0.02
Banana flour	116.45 ± 4.75	1.14 ± 0.00	1.46 ± 0.02
Soy protein isolate	261.26 ± 2.50	1.06 ± 0.00	2.72 ± 0.01
Egg white protein	70.19 ± 1.85	0.49 ± 0.00	0.11 ± 0.01
*Pasta samples*			
Semolina	55.73 ± 0.73 ^e^	0.10 ± 0.00 ^g^	0.31 ± 0.02 ^g^
Banana	63.37 ± 0.05 ^c^	1.05 ± 0.03 ^a^	1.04 ± 0.01 ^c^
BE5	54.66 ± 2.68 ^e^	1.02 ± 0.02 ^a^	1.01 ± 0.00 ^c^
BS5	63.11 ± 1.19 ^c^	0.89 ± 0.02 ^c^	0.95 ± 0.01 ^d^
BE10	58.47 ± 1.86 ^cde^	0.98 ± 0.00 ^ab^	0.90 ± 0.01 ^e^
BS10	79.78 ± 2.10 ^b^	0.98 ± 0.00 ^ab^	1.22 ± 0.01 ^b^
BE15	60.47 ± 1.77 ^cd^	0.94 ± 0.01 ^bc^	0.73 ± 0.01 ^f^
BS15	86.69 ± 2.86 ^a^	1.02 ± 0.01 ^a^	1.37 ± 0.02 ^a^
*General linear model with semolina pasta excluded from the calculation*			
*Type of protein*			
Egg white protein	59.24 ^b^	0.99 ^a^	0.92 ^b^
Soy protein isolate	73.24 ^a^	0.93 ^a^	1.15 ^a^
*Level of protein*			
0%	58.89 ^d^	1.03 ^a^	1.04 ^b^
5%	63.37 ^c^	0.96 ^b^	0.98 ^c^
10%	69.13 ^b^	0.98 ^b^	1.06 ^a^
15%	73.58 ^a^	0.98 ^b^	1.05 ^ab^

BE5, BE10, BE15: Pasta prepared from 100% banana flour with 5, 10, 15 g addition of egg white protein powder/100 g flour. BS5, BS10, BS15: Pasta prepared from 100% banana flour with 5, 10, 15 g addition of soy protein isolate/100 g flour. Results in the table represent the mean of triplicate measurements. Mean ± standard deviation. Values within a column within a group followed by the same superscript letter are not significantly different from each other (*p* > 0.05), according to Tukey’s test.

**Table 2 foods-09-00589-t002:** Protein content, protein digestibility and protein availability of cooked pasta.

Formulation	Protein	Protein Digestibility	Protein Availability
	% db	%	%
Semolina	12.26 ± 0.09 ^c^	85.81 ± 0.10 ^a^	10.52 ± 0.09 ^c^
Banana	3.88 ± 0.04 ^h^	69.04 ± 0.21 ^e^	2.68 ± 0.04 ^h^
BE5	7.49 ± 0.05 ^g^	72.72 ± 0.65 ^d^	5.44 ± 0.04 ^g^
BS5	8.06 ± 0.03 ^f^	72.54 ± 0.31 ^d^	5.85 ± 0.03 ^f^
BE10	10.83 ± 0.07 ^e^	74.77 ± 0.28 ^c^	8.10 ± 0.08 ^e^
BS10	11.39 ± 0.02 ^d^	76.88 ± 0.31 ^b^	8.76 ± 0.03 ^d^
BE15	14.36 ± 0.14 ^a^	77.67 ± 0.64 ^b^	11.16 ± 0.11 ^a^
BS15	13.87 ± 0.02 ^b^	77.67 ± 0.21 ^b^	10.77 ± 0.02 ^b^
General linear model with semolina pasta excluded from the calculation			
*Type of protein*			
Egg white protein	9.14 ^b^	73.55 ^b^	6.85 ^b^
Soy protein isolate	9.30 ^a^	74.03 ^a^	7.01 ^a^
*Level of protein*			
0%	3.88 ^d^	69.04 ^d^	2.68 ^d^
5%	7.77 ^c^	72.63 ^c^	5.64 ^c^
10%	11.11 ^b^	75.83 ^b^	8.43 ^b^
15%	14.12 ^a^	77.67 ^a^	10.96 ^a^

BE5, BE10, BE15: Pasta prepared from 100% banana flour with 5, 10, 15 g addition of egg white protein powder/100 g flour. BS5, BS10, BS15: Pasta prepared from 100% banana flour with 5, 10, 15 g addition of soy protein isolate/100 g flour. Results in the table represent the mean of triplicate measurements. Mean ± standard deviation. Values within a column within a group followed by the same superscript letter are not significantly different from each other (*p* > 0.05), according to Tukey’s test.

**Table 3 foods-09-00589-t003:** Comparison of amino acid profiles in samples pasta with daily adult intake recommendation (mg of amino acid per g protein).

Formulation	Amino Acid							
	His	Ile	Leu	Lys	Cys + Met	Phe + Tyr	Thr	Val
Semolina	57 ± 3 ^b^	24 ± 1 ^d^	53 ± 3 ^bc^	46 ± 2 ^d^	129 ± 7 ^a^	59 ± 3 ^c^	33 ± 2 ^cd^	29 ± 1 ^f^
Banana	137 ± 7 ^a^	7 ± 1 ^e^	23 ± 1 ^d^	127 ± 7 ^a^	2 ± 0 ^d^	32 ± 1 ^e^	64 ± 3 ^a^	14 ± 1 ^de^
BE5	60 ± 9 ^b^	31 ± 4 ^bc^	58 ± 4 ^bc^	77 ± 12 ^b^	71 ± 4 ^b^	58 ± 4 ^c^	44 ± 6 ^b^	40 ± 3 ^b^
BE10	53 ± 11 ^b^	40 ± 1 ^a^	71 ± 0 ^a^	78 ± 8 ^b^	76 ± 8 ^b^	75 ± 1 ^ab^	47 ± 4 ^b^	51 ± 1 ^a^
BE15	40 ± 3 ^b^	42 ± 1 ^a^	74 ± 3 ^a^	73 ± 3 ^bc^	73 ± 4 ^b^	78 ± 1 ^a^	44 ± 2 ^b^	52 ± 1 ^a^
BS5	42 ± 13 ^b^	26 ± 3 ^cd^	51 ± 4 ^c^	59 ± 8 ^cd^	45 ± 6 ^c^	48 ± 2 ^d^	30 ± 5 ^d^	28 ± 3 ^e^
BS10	62 ± 3 ^b^	31 ± 1 ^b^	60 ± 1 ^b^	80 ± 2 ^b^	70 ± 3 ^bc^	60 ± 1 ^c^	41 ± 2 ^bc^	34 ± 2 ^cd^
BS15	60 ± 2 ^b^	36± 1 ^ab^	69 ± 1 ^a^	84 ± 1 ^b^	107 ± 2 ^a^	69 ± 1 ^b^	42 ± 2 ^bc^	38 ± 1 ^bc^
EAA^*^	16	30	61	48	23	41	25	40
General linear model with semolina pasta excluded from the calculation								
*Type of protein*								
Egg white	72 ^a^	30 ^a^	56 ^a^	89 ^a^	55 ^a^	61 ^a^	50 ^a^	39 ^a^
Soy protein	75 ^a^	25 ^b^	51 ^b^	88 ^a^	56 ^a^	52 ^b^	44 ^b^	28 ^b^
*Level of protein*								
0	137 ^a^	7 ^d^	23 ^d^	127 ^a^	2 ^d^	32 ^d^	64 ^a^	14 ^c^
5	50 ^b^	28 ^c^	54 ^c^	68 ^b^	58 ^c^	53 ^c^	37 ^c^	34 ^b^
10	57 ^b^	35 ^b^	66 ^b^	79 ^b^	73 ^b^	68 ^b^	44 ^b^	42 ^a^
15	50 ^b^	39 ^a^	71 ^a^	79 ^b^	90 ^a^	73 ^a^	43 ^bc^	45 ^a^

* Recommendation of daily amino acid intake for adult human by WHO, FAO and UNU [26]. BE5, BE10, BE15: Pasta prepared from 100% banana flour with 5, 10, 15 g addition of egg white protein powder/100 g flour. BS5, BS10, BS15: Pasta prepared from 100% banana flour with 5, 10, 15 g addition of soy protein isolate/100 g flour. Results in the table represent the mean of triplicate measurements. Mean ± standard deviation. Values within a column within a group followed by the same superscript letter are not significantly different from each other (*p* > 0.05), according to Tukey’s test.

**Table 4 foods-09-00589-t004:** Amino acid scores (AAS) (mg per g protein) and protein digestibility corrected amino acid scores (PDCAAS) of sample pasta.

Formulation	Amino Acid								
	His	Ile	Leu	Lys	Cys + Met	Phe + Tyr	Thr	Val	PDCAAS
Semolina	3.57 ± 0.17 ^b^	0.80 ± 0.04 ^cd^	0.87 ± 0.04 ^bc^	0.96 ± 0.05 ^d^	5.62 ± 0.32 ^a^	1.44 ± 0.08 ^c^	1.31 ± 0.06 ^cd^	**0.73** ± 0.03 ^de^	0.63 ± 0.03 ^c^
Banana	8.57 ± 0.46 ^a^	0.22 ± 0.02 ^d^	0.38 ± 0.02 ^d^	2.65 ± 0.15 ^a^	**0.10** ± 0.01 ^d^	0.78 ± 0.03 ^e^	2.57 ± 0.12 ^a^	0.35 ± 0.02 ^f^	0.07 ± 0.01 ^e^
BE5	3.72 ± 0.15 ^b^	1.03 ± 0.05 ^bc^	0.94 ± 0.07 ^bc^	1.61 ± 0.25 ^b^	3.09 ± 0.18 ^b^	1.41 ± 0.10 ^c^	1.75 ± 0.26 ^b^	**1.01** ± 0.07 ^b^	0.69 ± 0.05 ^bc^
BE10	3.32 ± 0.70 ^b^	1.32 ± 0.02 ^a^	**1.16** ± 0.01 ^a^	1.63 ± 0.17 ^b^	3.29 ± 0.32 ^b^	1.82 ± 0.02 ^ab^	1.87 ± 0.15 ^b^	1.27 ± 0.02 ^a^	0.87 ± 0.00 ^a^
BE15	2.49 ± 0.20 ^b^	1.39 ± 0.03 ^a^	**1.21** ± 0.04 ^a^	1.52 ± 0.05 ^bc^	3.15 ± 0.17 ^b^	1.90 ± 0.07 ^a^	1.77 ± 0.08 ^b^	1.30 ± 0.03 ^a^	0.94 ± 0.03 ^a^
BS5	2.64 ± 0.80 ^b^	0.85 ± 0.11 ^c^	0.84 ± 0.07 ^c^	1.22 ± 0.17 ^cd^	1.94 ± 0.27 ^c^	1.17 ± 0.05 ^d^	1.20 ± 0.20 ^d^	**0.70** ± 0.08 ^e^	0.51 ± 0.06 ^d^
BS10	3.86 ± 0.19 ^b^	1.04 ± 0.05 ^b^	0.99 ± 0.02 ^b^	1.67 ± 0.06 ^b^	3.05 ± 0.12 ^bc^	1.47 ± 0.03 ^c^	1.65 ± 0.06 ^bc^	**0.84** ± 0.04 ^cd^	0.65 ± 0.03 ^bc^
BS15	3.72 ± 1.19 ^b^	1.20 ± 0.04 ^ab^	1.13 ± 0.01 ^a^	1.76 ± 0.03 ^b^	4.65 ± 0.94 ^a^	1.68 ± 0.01 ^b^	1.66 ± 0.06 ^bc^	**0.95** ± 0.03 ^bc^	0.74 ± 0.02 ^b^
General linear model with semolina pasta excluded from the calculation									
*Type of protein*									
Egg white	4.52 ^a^	0.99 ^a^	0.93 ^a^	1.85 ^a^	2.41 ^a^	1.48 ^a^	1.99 ^a^	0.98 ^a^	0.64 ^a^
Soy protein	4.70 ^a^	0.83 ^b^	0.84 ^b^	1.82 ^a^	2.43 ^a^	1.27 ^b^	1.78 ^b^	0.71 ^b^	0.49 ^b^
*Level of protein*									
0	8.57 ^a^	0.22 ^d^	0.38 ^d^	2.65 ^a^	0.10 ^d^	0.78 ^d^	2.57 ^a^	0.35 ^c^	0.07 ^d^
5	3.18 ^b^	0.94 ^c^	0.89 ^c^	1.42 ^b^	2.52 ^c^	1.29 ^c^	1.47 ^c^	0.85 ^b^	0.60 ^c^
10	3.59 ^b^	1.18 ^b^	1.08 ^b^	1.65 ^b^	3.17 ^b^	1.65 ^b^	1.76 ^b^	1.06 ^a^	0.76 ^b^
15	3.10 ^b^	1.29 ^a^	1.17 ^a^	1.64 ^b^	3.90 ^a^	1.79 ^a^	1.72 ^bc^	1.13 ^a^	0.84 ^a^

Numbers in bold show limiting value of AAS; BE5, BE10, BE15: Pasta prepared from 100% banana flour with 5, 10, 15 g addition of egg white protein powder/100 g flour. BS5, BS10, BS15: Pasta prepared from 100% banana flour with 5, 10, 15 g addition of soy protein isolate/100 g flour. Results in the table represent the mean of triplicate measurements. Mean ± standard deviation. Values within a column within a group followed by the same superscript letter are not significantly different from each other (*p* > 0.05), according to Tukey’s test.

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
