# Peer review of "Effect of Egg White Protein and Soy Protein Isolate Addition on Nutritional Properties and In-Vitro Digestibility of Gluten-Free Pasta Based on Banana Flour"

_foods, 2020, doi:10.3390/foods9050589_

Round 1

Reviewer 1 Report

The manuscript is prepared well, however I recomment to take into accound some comments listed below:

Introduction

The information about the content of main components of banana flour should be provided because the aim of the study is not clear. How much protein contain banana flour? There is Table 2 but in the Introduction should be some general idea included

Table 1 Too many digits after coma

Soy protein isolate is pH dependant. In this context the authors shoud add comments or some explanation why pH did not were adjusted when preparing pasta.

I recommend to add pictures of samples

Author Response

Thank you for your kind information and review.

PLease note the reply below.

Introduction

The information about the content of main components of banana flour should be provided because the aim of the study is not clear. How much protein contain banana flour? There is Table 2 but in the Introduction should be some general idea included

Thank you for your comments. I have added a little more information to the introduction. 

"Banana flour is mainly composed of starch, with a relatively low protein content, but the high level of bioactive ingredient in banana makes it useful as a functional foods ingredient."

Table 1 Too many digits after coma

Thank you, we have altered the table. 

Soy protein isolate is pH dependant. In this context the authors shoud add comments or some explanation why pH did not were adjusted when preparing pasta.

Added 

Thank you for all your comments.

Reviewer 2 Report

The manuscript deals with the nutritional characteristics of some gluten-free alternative foods. This line is of great interest for celiac people. The manuscript is well organized, clear and statistically tested. Only a few comments

pg 3. The equation for the protein digestibility requires a reference or the source.

pg 6. The sentence " there are also..." sounds rare.

pg 6. The sentence " This can also..." is not clear

Fig. 2. Please, at least revise the thickness of the lines to improve the plot

pg 8. The sentence "The results ..." could be simpler

pg 8. Please revise "Table 4..."

References. Please revise this section. At least some Journal names need revision.

Author Response

Thank you for your help in reviewing the manuscript and your comments. We have evaluated the comments and converted the manuscript as suggested. We feel that the paper is now much improved.

To be specific, please find the responses to your comments as below.

The manuscript deals with the nutritional characteristics of some gluten-free alternative foods. This line is of great interest for celiac people. The manuscript is well organized, clear and statistically tested.

Thank you for your kind comments, we appreciate your kind remarks.

Only a few comments

pg 3. The equation for the protein digestibility requires a reference or the source.

pg 6. The sentence " there are also..." sounds rare.

Altered as suggested.

pg 6. The sentence " This can also..." is not clear

Sentence altered, apologies for the misunderstanding.

Fig. 2. Please, at least revise the thickness of the lines to improve the plot

pg 8. The sentence "The results ..." could be simpler

We have revised the sentence

pg 8. Please revise "Table 4..."

We have revised the sentence

References. Please revise this section. At least some Journal names need revision.

We have tried to revise the references